# Fungicide Scent Pollution Disrupts Floral Search-and-Selection in the Bumblebee *Bombus impatiens*

**Nour Yousry, Paige Henderson and Jordanna Sprayberry ***

Departments of Biology & Neuroscience, Muhlenberg College, Allentown, PA 18104, USA
* Correspondence: jordannasprayberry@muhlenberg.edu

**Abstract:** Bumblebees are valuable generalist pollinators. However, micro- and macro-stressors on bumblebees negatively impact both foraging efficiency and pollination efficacy. Given that colonies have a resource threshold for successful reproduction, factors that decrease foraging efficiency could negatively impact conservation efforts. Recently, agrochemical odor pollution has been shown to hinder floral odor learning and recognition in *Bombus impatiens* via an associative odor learning assay (FMPER). These results may have implications for the field foraging behavior of bumblebees. Building on this prior work, our study aimed to determine if negative effects of fungicides on associative odor learning and recognition scale up to negative impacts on actively foraging bumblebees. These experiments investigated whether the presence of a background fungicide odor (Reliant® Systemic Fungicide) impacts the location of a learned floral resource (lily of the valley-scented blue flowers) in a wind tunnel. Experiments were run with and without early access to visual cues to determine if fungicide odor pollution is more impactful on bees that are engaged in olfactory versus visual navigation. Fungicide odor pollution reduced landing frequency in both paradigms.

**Keywords:** bumblebee; olfaction; foraging; fungicide

## 1. Introduction

### 1.1. Bumblebees Serve as Vital Insect Pollinators in Our Ecosystem

Demands for increased produce output are expected to rise about 1.2% in the agricultural industry annually; this call for higher agricultural yield emphasizes the need for robust pollinator populations. The global agricultural industry currently relies on the work of pollinators, so much so that over a third of all crops grown and three quarters of all angiosperms require pollinators to maintain their species [1]. Fruits and vegetables, among other crops, rely heavily on animal pollinators; with pollinators directly contributing to the successful reproduction of 87 crops worldwide, with Western honey bees (*Apis mellifera* L.) and bumblebees (*Bombus*) serving as the main contributors to global pollination efforts and crop quality [2,3].

### 1.2. Agrochemicals Are a Contributing Stressor in Pollinator Declines

Unfortunately, bumblebees and other pollinators are experiencing alarming declines [4–6]. Bee populations face many macro- and micro-scale stressors, including: rapidly changing climate [7,8], lack of habitat conservation efforts [9,10], increases in diesel exhaust pollution [11], pathogen transmission [12,13] and pesticide exposure [14–19]. Despite ongoing population declines, agrochemical use is increasing because of efforts to maximize crop yields, due to the industry's heavy reliance on these chemicals (including pesticides, fungicides, and other insecticides) [20]. Prior research demonstrates that pesticides have made their way into the tissues of bumblebees, with neonicotinoid insecticides capable of leaving residues in bees within a few minutes of exposure during foraging [21]. While certification or licensing of agrochemicals by the Federal Drug Administration (FDA) and the Medicines and Healthcare Regulatory Agency (MHRA) requires demonstration of 'bee-safety', these

application guidelines are based on LD50 tests rather than behavioral metrics; moreover, these agrochemicals are tested and certified in isolation despite the fact that agrochemical mixtures have been shown to have negative impacts [22]. However, sublethal doses of some pesticides, such as neonicotinoids, exert problematic effects via behavioral modification. For example, bumblebees have been shown to collect less pollen, learn floral cues and tasks slower, and demonstrate a lack of foraging motivation when exposed to neonicotinoid pesticides [22–24]. Field realistic exposure to sulfoximine insecticides reduces bumblebee foraging activity [25]. Furthermore, insecticides have been shown to have an effect on colony nectar storage and the queen bee's foraging efforts [26].

### 1.3. Agrochemicals Indirectly Disrupt Bumblebee Learning and Foraging Behavior

In addition to the consequences of direct exposure on behavior detailed above, agrochemicals can also modify behavior through indirect disruption of floral-sensory cues. Negative effects of odor pollution have been previously demonstrated in relation to air pollution modifying the structure of floral odor plumes [27–29]. However indirect effects are not limited to nitrous oxides and ozone degradation of odor plumes, as agrochemicals themselves can have strong scent signatures that additively modify floral odor information. Existing work has convincingly shown that agrochemical odor pollution is disruptive to bumblebee foraging behavior in lab contexts [30,31]. Early work found that a background of fungicide odor reduced successful navigation through a walking maze and increased the time it took for bees to locate a scented feeder within that maze. The same study showed that bumblebees preferred unpolluted foraging chambers as compared to fungicide- and fertilizer-odor contaminated chambers [30]. These findings could result from several possible mechanisms of disruption. First, the pollution background could have decreased the contrast of the learned odor relative to the environment. Second, the pollution background could have combined with the learned odor and caused a shift in odor blend structure. Finally, the odor pollution could have had a valence-based effect, where aversive components of pollution scent pushed back against attraction to the learned (appetitive) odor.

A follow-up study used an associative odor learning paradigm (free moving proboscis extension reflex, or FMPER) to examine the impact of different pollution modalities on floral odor learning and recognition, specifically, testing background odor pollution during learning and recognition of floral scent, background odor pollution during recognition of (an already learned) floral scent, and point pollution presented in contrast to the floral scent during recognition. Three different fungicides (Safer, Scotts, and Reliant) were tested across all pollution modalities. All fungicides disrupted floral odor recognition in all pollution modalities, although patterns of response across fungicide concentration were not identical [31]. All three fungicides tested had angular distances from the learned floral odor (*Monarda fistulosa*) that were well above the discrimination threshold for bumblebees [32], so reduced contrast between floral odor and pollution background is not likely driving disruption. However, the differences in patterns of disruption across fungicides imply that multiple mechanisms may be at play. Scotts was disruptive at all concentrations and in all paradigms, while Reliant and Safer typically induced stronger disruption at lower concentrations. Taken together this implies that Scotts is likely aversive (a valence-based effect), while Reliant and Safer are not, based on findings that higher concentrations of these two fungicides did not disrupt responses to *M. fistulosa* scent. Their capacity to disrupt scent recognition at lower concentrations suggests that when odors are at the threshold of perception, or liminally perceived, neural responses to fungicide odor may not be strong enough to encode a separate odor object and may instead modify responses to floral odor, resulting in an encoding shift. These data also imply that if neural identity of floral odor is shifted by liminal odor pollution, the 'new' odor is not readily generalized to the original. With loss of generalization, we can hypothesize that when testing the effects of odor pollution on foraging bumblebees we will see a disruption of odor-driven foraging behavior. To tackle this experimentally, we need to acknowledge that foraging behavior is not monolithic; rather, it is a constellation of sub-behaviors. Recent work proposed

a framework for identifying and understanding foraging phases in terms of state and state-transitions (search (S), acquisition (A), navigation (N), S→A, N→A, etc.), forager-background (naive (n), experienced (e), primed (p)), and spatial scale (local (l), intermediate (i), distant (d)), with the understanding that different types of sensory information are relevant to different foraging phases [33]. The effects of agrochemical odor pollution are most likely to manifest in phases that utilize odor information, including, but not limited to: (1) search-to-navigation motivated by odor encounter at intermediate and local spatial scales (S→N(i,n/e/p); S→N(l, n/e/p)); (2) navigation at intermediate scales (N(i, n/e/p)); and (3) search-to-acquisition at local spatial scales (S→A(l, n/e/p)). Search-to-navigation represents scenarios where bumblebees are searching for novel resources in an environment and encounter a floral odor plume in isolation (an intermediate spatial scale) or in conjunction with a visual target (a local spatial scale), triggering either odor-guided navigation (N(i,n/e/p)) or visually-guided (N(l,n/e/p)) navigation. Sensory-guided navigation will become a local search once animals are in the vicinity of and receiving sensory cues provided by resolvable flowers (i.e., flower shape, nectar guides, scent marks). The choice to land on a flower is the search-to-acquisition transition. The experiments presented in this manuscript: (1) determine if previously tested fungicide-odors are consistently disruptive to floral scent recognition, using recognition of lily of the valley (LoV) scent in FMPER tests; and (2) bring this fungicide work into a foraging behavior context by testing for effects of fungicide odor pollution on sensory-guided navigation towards a learned cue (N(i/l,e)) and subsequent search-acquisition likelihood (S→A(l,e)) via a wind tunnel paradigm.

## 2. Materials and Methods

### 2.1. Animals

*Bombus impatiens* colonies consisting of 100–125 bees, supplied by Koppert Biological Systems (Howell, MI, USA) and Biobest Sustainable Crop Management (Romulus, MI, USA), were maintained in a lab environment. Each colony was wrapped in a seedling heat mat (IPower Propagate, China) with a thermostat to ensure a consistent temperature range of 75–85 °F in the hive. These experiments used 6 colonies from May 2022 to February 2023. To increase motivation to participate in wind-tunnel experiments, bumblebees were limited in their sucrose access and given a 2–3 h feeding period each day. Bumblebees used in FMPER experiments were given ad libitum access to 30% sucrose. All bees had ad libitum access to ground pollen.

### 2.2. Free-Moving Proboscis Extension Reflex (FMPER) Protocol

FMPER Setup: Associative odor (AO) learning was measured using a modified Free Moving Proboscis Extension Reflex (FMPER) [31,32,34]. Healthy, active-individual *B. impatiens* (from Koppert Biological Systems) were selected from lab colonies, placed in screen-backed vials, acclimated for 2 h, and placed into an odor stimulation apparatus (Figure 1a). The ventilating testing array drew air in through two small holes in the lid and out the back, with flow rates ranging between 0.1 to 0.3 m/s (VWR-21800-024 hot wire anemometer). During conditioning, bees were offered a single drop of 50% sucrose on a blue strip inserted through one of the two lid holes (hole selection was randomized). These strips were cut from plastic folders and had absorbent adhesive bandage tape (Cover Roll) placed on the back to hold a 1 μL lily of the valley (LoV) odor stimulus. The plastic prevented the odor solution from diffusing into the sugar solution on the top of the strip; therefore, the primary sensory encounter with odorants was through the olfactory rather than the gustatory system. Bumblebees would undergo four association trials at 5 min intertrial intervals, during which they were presented LoV paired with sucrose reward via the blue plastic strip. Bumblebees that successfully completed four association trials would then undergo a test trial after an additional 5 min wait, where they were presented with two unrewarding blue plastic strips, one containing LoV stimulus and one containing unscented mineral oil (MO) (Figure 1b). Proboscis extension on the LoV strip was considered a "correct" choice,

on the MO strip was considered an "incorrect" choice, and individuals that approached the strips three times without exhibiting PER or reacting to odor presentation within 30–45 s were classified as "no-choice". All tested bees were tagged after experiments to prevent re-testing, ensuring statistical independence of data points. If all bees tested on a given day for a given stimulus (i.e., an experimental session) responded "no choice", that session's data was excluded from the final dataset to protect against a potential experimenter error disrupting the dataset. Across the 176 bees tested from five colonies, nine data points (5.11%) fell into this exclusion category (Supplemental Dataset S1).

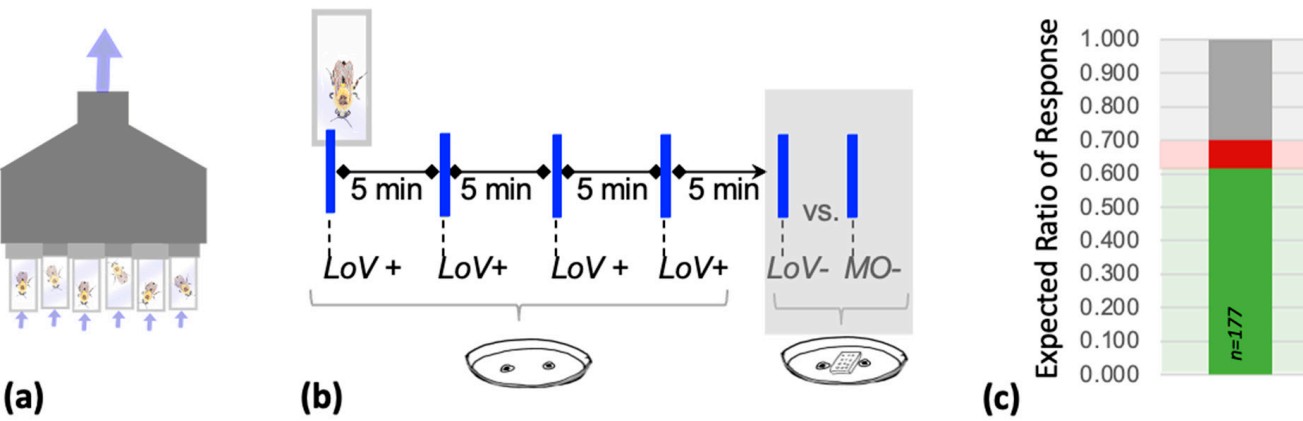

**Figure 1.** (**a**) The FMPER rig holds 6 bees in ventilating manifold that draws air through individual tubes. (**b**) Bumble bees undergo 4 conditioning trials at five minute intervals, where blue strips scented with Lily of the Valley (LoV) are paired with a 50% sucrose reward (+). Before the test trial the regular lid is removed, and replaced with a lid containing a mesh cage. For control trials, caps without a mesh cage were used, while pollution trials contained a fungicide in the lid's mesh cage, creating a background fungicide odor during floral-scent testing. Test trials presented bees with two unrewarding strips (−), one LoV-scented and one with unscented mineral oil (MO). (**c**) The expected distribution of responses in FMPER odor-learning experiments (established in Sprayberry 2020). The data shown are from 177 bees trained with 7 different associative odors tested against MO. Green represents a correct choice, red an incorrect choice, and grey no choice.

Agrochemical Testing: Training trials, which confirmed the ability of bumblebees to learn LoV, had no pollution present during conditioning and test trials. To mimic the effects of fungicide being applied to an entire plant, creating a background of polluting odor, we added a 3D-printed box with a perforated top holding 1 µL of fungicide on filter paper to the inside of vial lids (see for [31] details). To test the effects of background odor pollution on LoV recognition, the box would be placed at the start of the 5 min wait before testing trials (Figure 1b). These experiments tested three different environmentally relevant fungicides: Safer R© Brand Garden Fungicide II (active ingredient: sulfur, 0.4%), Scotts R© Lawn Fungus Control (active ingredient: thiophanate-methyl, 2.3%), and Reliant R© Systemic Fungicide (Agri-Fos/Garden-Phos) (active ingredient: mono and dipotassium salts, 45.8%). Two of the fungicides, Safer R© Brand Garden Fungicide II and Reliant R© Systemic Fungicide (Agri-Fos/Garden-Phos), are water-soluble and come in liquid formulations. As in David et al., Safer R© was used undiluted to maintain the active ingredient compositional makeup, and Reliant R© was prepared at a 1:100 dilution to match the concentration of active ingredient found in the Safer R© fungicide (0.4%) [31]. The Scotts R© Lawn Fungus Control came in solid (granule) formulation and required water to activate ("Lawn Fungus Control", "Safer R© Brand Garden Fungicide II"), as per the product label. To prepare a liquid solution, granules were mixed with deionized water in a 1:6 volumetric ratio of fungicide:deionized water and allowed to sit overnight. Fungicide solutions were disposed of after 72 h.

### 2.3. Wind Tunnel Experiments

Association: As previously mentioned, wind tunnel colonies had restricted sucrose-access. During timed feeding sessions, a glass feeder with an LoV-scented blue 3D-printed flower/s (one or two) containing 30% sucrose was placed in the foraging chamber. Scent-stimuli were 5 uL LoV pipetted onto a 2 cm × 2 cm piece of absorbent tape (Cover Bandage) on the back of each flower, if only one flower was used 10 uL of LoV was pipetted. Colonies were eligible for testing after three consecutive days of flower color and scent exposure. Occasionally, throughout a colony's life span, foraging activity within this window was not robust enough to maintain adequate levels of honey for the hive. In these cases, ad libitum feeding was provided from a glass feeder with no floral stimuli for two to three days. After this, colonies would have an additional two days to re-associate with color and scent stimuli. Following timed feeding sessions, 3D-printed flowers were washed, dried, and aired out overnight. Sucrose feeders were also washed and refilled if any sucrose discoloration or fungal growth was observed.

Wind Tunnel Structure: The wind tunnel dimensions were 5.625 ft × 1.625 ft × 1.541 ft; the walls and floor were white corrugated plastic and the ceiling was transparent plexiglass. Green tape ran lengthwise on the walls, floor, and ceiling to provide visual cues of the wall locations without providing wide field motion cues for distance measurement by flying bees [35]. Air flow through the tunnel was provided by a box fan, with insulation and grid sheets to even out flow. Panels were placed at $^1/_3$ and $^2/_3$ of the way through the tunnel on opposite sides, making a serpentine maze (Figure 2). Air flow through the tunnel is not laminar, but glycerin smoke plumes were visualized to confirm that scent plumes are carried all the way through the tunnel. The average air flow sampled in the center of the tunnel in each section was 0.37 ± 0.11 m/s. Test bees were introduced into the tunnel at a consistent location, via a launch platform that accepts bee vials. Test flowers were placed at the opposite end of the tunnel, 1.5 m (59 in) from the launch platform. Test flowers were unrewarded copies of training flowers, scented with 10 uL of 1:100 LoV.

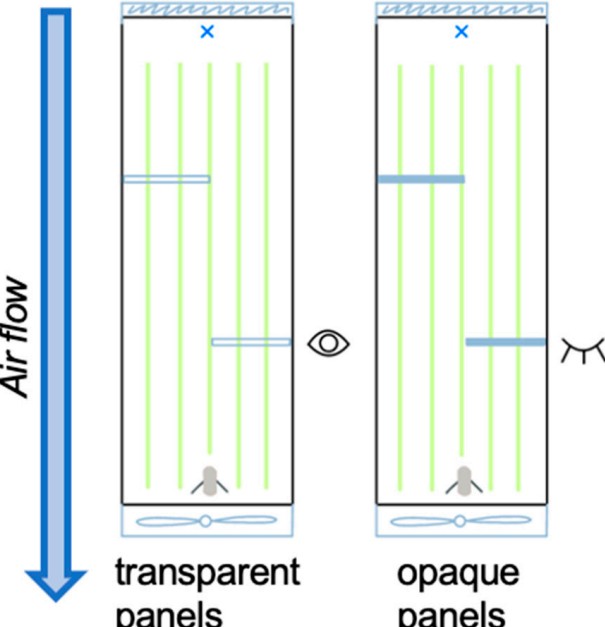

**Figure 2.** The three wind tunnel configurations used in these experiments. The "x" represents the flower location, while the bee represents the launch tube location. In the first two configurations panels were used to create a 'maze'. Transparent panels allowed bees in the launch tube to see flowers, mimicking a local spatial scale. Opaque panels hid visual cues; access to odor cues then mimicking a distant spatial scale.

Testing: Bees that were actively feeding from training flowers were captured in plastic vials for wind tunnel experimentation. Bees waited in their vials on a heating mat (80.6–82.4 °F) for 5–30 min before being placed into the wind tunnel on the launch platform. The cap to a bee's vial was removed to allow the bee to exit; bees that did not exit their vial within five minutes were excluded. After exiting the vial, bees were given 10 min to locate/land on the test flower. Bees that landed on the test flower were marked as "land"; bees that did not land after 10 min were removed from the wind tunnel. Experimental days where no bees landed were excluded from the final dataset to protect against potential experimenter error disrupting the dataset. Across the 152 bees tested from 7 colonies, 27 data points (17.76%) fell into this exclusion category (Supplemental Dataset S2). All tested bees were tagged after experiments to prevent re-testing, ensuring statistical independence of data points.

### 2.4. Experimental Conditions

Experimental Conditions/Testing Cue Encounter: Previous work indicates that searching bumblebees most likely encounter floral cues either by odor first, or by odor and visual cues simultaneously [36]. We used transparent plexiglass panels to allow bees to see and smell-test flowers at launch, allowing for a simultaneous cue encounter and mimicking 'floral selection'. We used opaque white plastic panels to prevent bees in the launch tube from seeing the test flower at launch, creating a scenario where searching bees encounter odor in isolation and mimic 'floral search' (Figure 2).

Experimental Conditions/Effects of Reliant® Fungicide Odor Pollution: To assess the effects of background odor pollution on bumblebees' navigation towards and landing frequencies on learned flowers, a 10 in × 6.5 in gridded plastic mesh tray with 5 × 150 mm-diameter filter-paper rounds was mounted on the back screen of the wind tunnel, behind the scented flower. Control trials had unaltered filter paper, while test trials used 14.1 mL of Reliant® Systemic Fungicide. This is equivalent to 0.477 fluid ounces (0.00373 gallons), which is a field-realistic dose for the square footage of the wind tunnel's back wall. Of the three fungicides tested in FMPER trials, only Reliant had manufacturer's instructions that were tractable with wind tunnel testing. Separate mesh trays used for control and Reliant® Systemic Fungicide were not interchanged to avoid any cross contamination. Mesh trays were placed in the fume hood overnight after experiments to allow for the evaporation of any residual volatiles.

### 2.5. Statistical Analysis

FMPER: To confirm that *B. impatiens* are capable of learning LoV, as has been previously demonstrated [32], we compared the distribution of "correct" (C), "incorrect" (I), and "no choice" (NC) for control trials to a random distribution (1:1:1) with an exact goodness of fit test using a log likelihood ratio method for calculating $p$-values ($p < 0.001$). Rejection of the null-hypothesis in this test would indicate a non-random distribution of responses. Next, we tested the control data against an expected response distribution for successful odor learning and recognition in this FMPER paradigm. This distribution (C = 61.6%, I = 8.5%, NC = 29.9%) is based on FMPER data from 177 bees using 7 different AO stimuli tested against unscented mineral oil—representing normal associative odor learning behavior in this paradigm; full details are available in Sprayberry (2020). Large $p$-values, or acceptance of the null hypothesis, in this test would indicate that B. impatiens' responsiveness to LoV is 'normal' and not disrupted. Following the recommendations of Amrhein et al., we are reporting exact $p$-values and not classifying data into binary categories of "significant" versus "insignificant" [37]. However, readers wanting to assess the traditional significance of $p$-values for this particular data set will want to use the Bonferroni-corrected alpha value of 0.025 for the control data, and an alpha value of 0.05 for the remaining datasets. Datasets with $p$-values less than 0.15 were subjected to a binomial post hoc comparison to theoretical distribution for the "correct", "incorrect", and "no choice" data. Datasets from pollution FMPER experiments were analyzed against the expected response distribution using the exact goodness of fit test to determine if learning of/responsiveness to LoV is impaired

by background fungicide odor. Small *p*-values, or rejection of the null hypothesis, would indicate disruption.

Wind Tunnel: Wind tunnel statistical analysis asks: do *B. impatiens* select a learned cue in a free flight environment? For this, we compared the distribution of "landed" and "didn't land" when pollution was or was not present in different wind tunnel set ups. In comparing these two choices, a Fisher's exact test was used to compare "landed"/"didn't land" ratios between unpolluted and polluted conditions. This test was used to calculate *p*-values and was repeated across all wind tunnel conditions (opaque panels and clear panels). For animals that successfully landed, time to flower was recorded. The average time to flower for control versus polluted conditions were compared via a *t*-test for each wind tunnel configuration.

## 3. Results

### 3.1. Fungicide Odor Pollution Reduces Bumblebee Responses to a Learned Floral Odor, Lily of the Valley, in an Associative Odor Learning Paradigm

We verified that bumblebees were able to associate an LoV odor signal with a sucrose reward (Figure 3). The FMPER response distribution (56% correct, 44% no choice) was different from random chance ($p < 0.001$) but was not different from the expected % correct response in post hoc binomial comparisons ($p = 0.34$, Table 1). Background fungicide odor pollution was introduced during testing trials to explore its effects on responses to the learned LoV cue. Across all tests it was observed that fungicide scents disrupted learned odor responses, as seen by a decrease in % correct and an increase in % no-choice (Figure 3), indicated by post hoc binomial comparisons (Table 1).

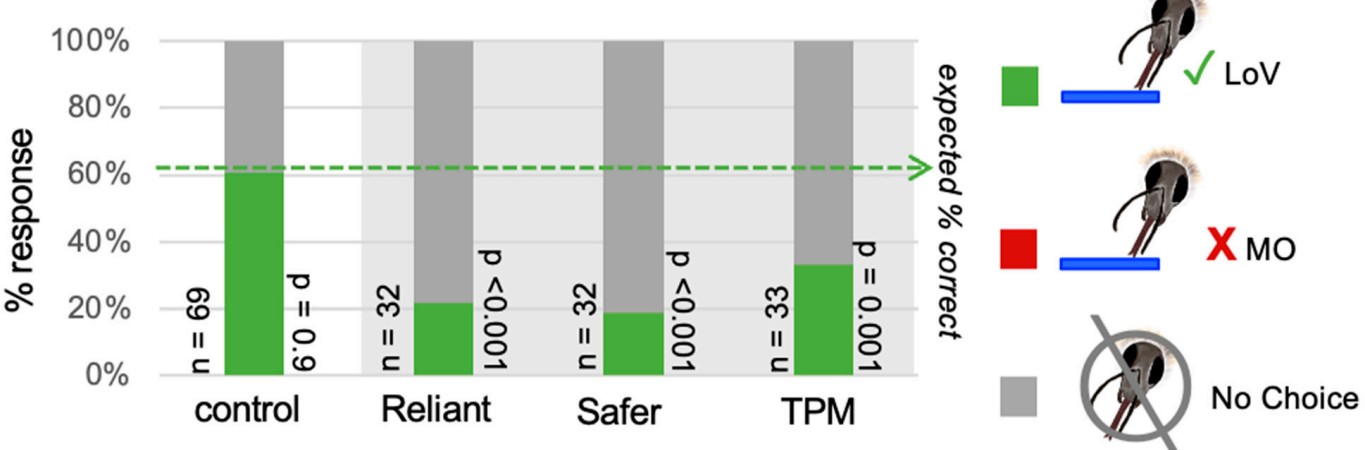

**Figure 3.** FMPER results indicate that odor pollution from all three fungicides disrupts odor responsiveness to learned lily-of-the-valley odor. The sample size for each test is listed to the left of each bar, while the *p*-value for %correct against expected is listed on the right side of the bar.

**Table 1.** FMPER results and statistics.

| Fungicide | %C/I/NR | Sample Size | *p*-Value (Compared to Expected) | Post Hoc: Correct *p*-Value | Post Hoc: Incorrect *p*-Value | Post Hoc: No Choice *p*-Value |
|---|---|---|---|---|---|---|
| Control (no fungicide) | 61/0/39 | 75 | 0.002 | 0.9 | 0.004 | 0.11 |
| Reliant | 22/0/78 | 32 | <0.001 | <0.001 | 0.11 | <0.001 |
| Safer | 19/0/81 | 35 | <0.001 | <0.001 | 0.11 | <0.001 |
| Scotts | 33/0/67 | 33 | <0.001 | 0.001 | 0.11 | <0.001 |

### 3.2. Fungicide Odor Pollution Reduces Bumblebee Landing Frequencies in Selection-and-Search Paradigms

Trials in the clear paneled maze allowed bees to visualize the flower upon launch, mimicking floral selection. In these tests, Reliant® odor pollution decreased bumblebees' landing frequency (21% as compared to 54% in controls, *p* = 0.026, Fisher's exact test) (Figure 4). Trials in the opaque-paneled maze provide only odor cues at the launch tube and mimic floral search. These tests also showed that Reliant® odor pollution decreased bumblebees' landing frequency (29% as compared to 50% in controls, *p* = 0.034, Fisher's exact test) (Figure 4). Fungicide odor pollution did not impact average time to flower (Table 2).

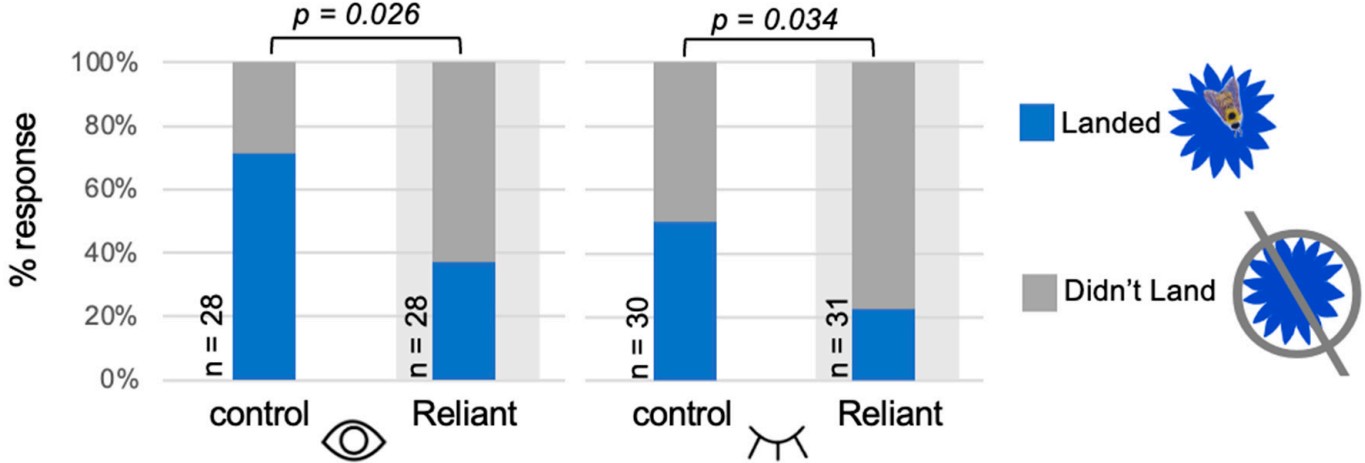

**Figure 4.** Paneled wind tunnel results indicate that fungicide odor-pollution reduced the likelihood of bumblebees landing on a learned floral cue. The sample size for each test is listed to the left of each bar, while the *p*-value for comparison between treatments is provided at the top of each plot. Trials run with transparent panels are shown on the left, those with opaque panels are on the right.

**Table 2.** Time to flower data from wind tunnel experiments.

| Wind Tunnel Configuration | Treatment | Average Time to Flower (min) | n (Bees That Landed) | *p* Value (*t*-Test) |
|---|---|---|---|---|
| Transparent | Control | 3.11 | 15 | 0.75 |
|  | Reliant | 3.49 | 6 |  |
| Opaque | Control | 2.6 | 15 | 0.55 |
|  | Reliant | 3.42 | 7 |  |

## 4. Discussion

### 4.1. Foraging Behavior and Conditioned Odor Recognition Are Hindered by Fungicide Odor Pollution

Agrochemicals have been shown to hinder learned odor behavior in bumblebees and honeybees with both direct exposure [18,22,38,39] and indirectly through odor pollution [30,31]. Commensurate with David et al.'s findings for *Monarda fistulosa* odor [31], the FMPER data here shows that three different fungicides, Safer R© Brand Garden Fungicide II, Scotts R© Lawn Fungus Control, and Reliant R© Systemic Fungicide (Agri-Fos/Garden-Phos) decrease responsiveness of bumblebees to learned lily of the valley (LoV) odor as indicated by a decrease in % correct responses and an increase in % no-choice. Interestingly, bumblebees made no incorrect responses, implying that the change in responsiveness is not due to an inability to perceive LoV odor during the test phase. While these associative odor learning experiments are valuable exploratory tools, they do not account for the animal state. Recent data on the neural substrates of selective attention in insects implies that identical sensory stimuli are not processed identically over time [40]. Likewise, studies showing that multisensory information can set a context for behavioral choice [41] implies that the 'state' of an insect is a critical component of how that animal will process sensory

information—a paradigm that is gaining more traction in insect sensory studies [42]. In an effort to test bumblebees in a physiologically relevant state, we tested the effects of odor pollution on freely flying and foraging animals via a wind-tunnel assay. We varied the availability of sensory information to bees at the wind tunnels launch site, endeavoring to set a physiological context that matched ethologically relevant foraging phases; with visual + olfactory availability mimicking sensory navigation at local spatial scales (N(l,e)), and olfactory availability mimicking odor navigation at intermediate spatial scales (N(i,e)). These phases precede floral selection, or search-to-acquisition, which we measured as flower landing. Our data indicate that odor pollution from Reliant© is disruptive in both paradigms, reducing landing frequency (Figure 4). Both Reliant and lily of the valley odors have been characterized in previous studies using a quantification method (Compounds Without Borders, or CWB) that vectorizes complex odors into dimensions representing a functional groups and carbon characteristics of their volatile components [31,32] (vectors available in Supplementary Materials S3). A heatmap comparing their vectors shows that both odors contain similar power in aromatics and six-carbon cyclic structures, but little else (Figure 5). Reliant has more power in methyls and oximes, while lily of the valley has more alcohol and aldehyde. Reliant also has a broader distribution of power across carbon chain lengths, while lily of the vally is concentrated in two dimensions: 3- and 8-carbon chain lengths.

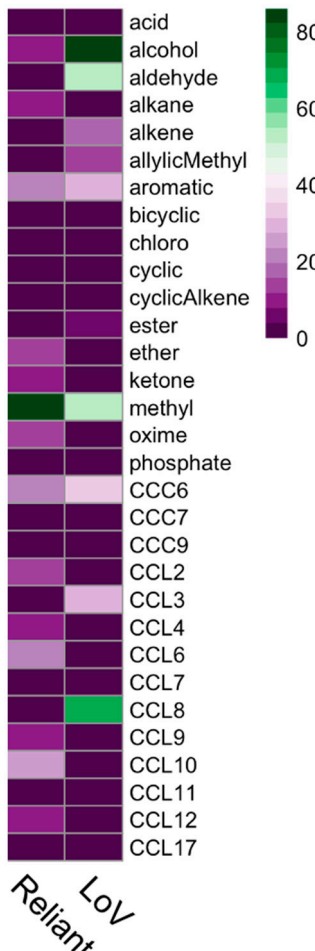

**Figure 5.** Heatmaps of the non-zero dimensions from Reliant and lily of the valley odor vectors show similarities and differences in odor structure. Odor vectors used are from previously published work (Sprayberry 2020 [32] and David et al. 2022 [31]).

### 4.2. Limitations and Implications

An alarming recent study by Ryalls et al. field tested the effects of subtractive odor pollution on pollination services [29]. Laboratory work had demonstrated that air pollutants, such as nitrogen oxides and ozone, degrade floral scents and disrupt pollinator odor responses [11,27,43–46]. Field tests later found that nitrogen oxides and ozone reduced pollinator floral visits by 83–90%. This study measured activity of a broad pollinator species range. From the perspective of bumblebees, these findings are most germane to experienced, returning foragers ($N(d/l,e)$), because the study design is spatially constrained to single patches and it has been well established that bumblebees rapidly configure regular routes to known resources [47–51]. This context makes our current findings all the more alarming. Bumblebees forage over spatial ranges up to or exceeding 1 km from their nest site [52,53]. If, as the Ryalls et al. study suggests, subtractive odor pollution causes bumblebees to abandon a known resource, they will need to search for novel replacement patches. Our findings imply that agrochemical odors could dissuade searching bumblebees from selecting scent-contaminated patches. Moreover, the use of diesel farm equipment means that subtractive and additive odor pollution are likely both occurring in many agricultural settings, indicating a need for integrative studies. Indeed, field work is critically important to fully understand the potential for agrochemical odor pollution to disrupt pollination services.

**Supplementary Materials:** The following supporting information can be downloaded at: https://www.mdpi.com/article/10.3390/agrochemicals2020013/s1, File S1: FMPER Data; File S2: Wind Tunnel Data. File S3: Odor Vectors.

**Author Contributions:** Conceptualization, N.Y. and J.S.; methodology, N.Y., P.H. and J.S.; validation, N.Y., P.H. and J.S.; formal analysis, J.S.; investigation, N.Y., P.H. and J.S. resources, J.S.; data curation, N.Y., P.H. and J.S.; writing—original draft preparation, N.Y. and P.H.; writing—review and editing, J.S.; visualization, N.Y. and J.S.; supervision, J.S.; project administration, J.S. All authors have read and agreed to the published version of the manuscript.

**Funding:** This research received no external funding.

**Institutional Review Board Statement:** Not applicable.

**Informed Consent Statement:** Not applicable.

**Data Availability Statement:** All data are available for download from the Supplementary Materials for this manuscript.

**Acknowledgments:** We thank Muhlenberg College for providing funding for this research via the Crist and Vaughan summer fellowships.

**Conflicts of Interest:** The authors declare no conflict of interest.

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
