# Peer review of "Fungicide Scent Pollution Disrupts Floral Search-and-Selection in the Bumblebee Bombus impatiens"

_agrochemicals, doi:10.3390/agrochemicals2020013_

Round 1
Reviewer 1 Report
At the heart of this manuscript is a really nice study that advances what is known about pollution's effects on floral choice at a seldom explored level of behavioral observation. This is an important level to explore and the authors lay out a nice set of wind tunnel experiments to not only examine the effects of pollution on behavior but attempt to discern visual and olifactory response based on their wind tunnel layout. These are undoubtedly challenging experiments to carry out and their results are quite convincing. I am grateful for this work being pursued.
The current organization of the manuscript is challenging to read because it really only lays out the objectives for the wind tunnel experiments but packs in several other objectives that are not well introduced, or not introduced at all. Either a slimmer manuscipt that sticks more closely to the wind tunnel experiments or a thicker manuscript that provides more perspective on the other objectives would greatly improve what the authors are trying to convey.
The abstract only lists one type of experiment and conclusion (the wind tunnel experiment) and that is the only aspect included in the introduction, other than mentioning that they confirm prior results from FMPER testing. It is unclear why the FMPER testing needed to be done, based on the introduction. The FMPER testing makes perfect sense as an initial smaller scale behavioral response to see if the authors can detect an effect of pollution in that kind of experiment, and then to step up from that to actual foraging to see if the effect remains. If that is how the authors see it, it would help to introduce it that way.
The most confusing part of the manuscript was Figure 1. RIght after a description of the FMPER setup in this manuscript in which blue plastic strips were used for odor cues there is a figure describing FMPER tests with yellow plastic strips. It took me awhile to realize that this is a figure from another study (it doesn't say so in the figure caption) and there is no reason at that point in the manuscript to expect data from another study being critical to analyzing this one. The mystery didn't unravel at all until I saw in the stats section that the data from a prior study was being used to generate expected response distributions for FMPER in the current study. But even that was confusing because it wasn't laid out why this was being done. I simply expected to see the kind of analyses comparing responses between control and pollution trials that appear at the beginning of the results section. What is the goal of the comparison with the expectations from the other paper? Is it testing the predictive power of the model from that paper? Giving more statistical power to the ones in this paper? The expectated correct response of bees to choice tests in this paper was 61.6%, based on the other paper. In the other paper that is an expected successful choice response when two compounds have lower contrast. The Lily of the Valley against mineral oil is high contrast in both studies, so is the test using that expected positive response rate asking if the polluted LoL is similar enough to LoL to get as many positive responses as a lower contrast choice, or am I misunderstanding how the other paper is being used in this comparison? There is a lot of really interesting insights in the other paper --I am simply unclear on how it is being used in the current paper, especially given that the chemical structures of LoL and the pollutants are not being compared in this paper. So I'm truly not sure if the references to the other paper are related to the primary wind tunnel objectives or if they represent a test of unstated objectives in the current paper.
Two questions from the wind tunnel experiments. Wasn't there a test without panels? I didn't see results for this. Also, what was the idea behind the 'noisy cue' experiment? It wasn't introduced and really wasn't interpreted.
The discussion is too brief. Should break out a paragraph on the insights from the FMPER part of the study. I also think you sell yourself short in the last paragraph, especially in reference to the Ryalls paper. It is a very good paper. Despite being a field study, it is still has unrealistically small plots for the context of foragers in a polluted landscape and it doesn't have the behavioral observations of yours. Your study and that one together make a nice and needed next level of ecological realism from vial choice tests and theoretical studies. You are making nice progress in this hard to study field.
Some smaller points:
The words modulate and modulation: I think modify and modification is meant in all cases where modulate and modulation are used
Bee willingness: This comes up several times. The word implies something cognitive beyond just a description of what the bees did or didn't do, but there is no evidence that justifies its use with bees. Lars Chitka's lab may yet prove bees have free will in addition to playfulness and social learning and culture, and when he does I'll take back my hesitation :)
Control trials: This comes up a number of times, as in line 114, when the term training trials would be clearer, since it isn't a control for the pesticides introduced later.
Line 166-167: odd sentence. Is it supposed to be "bees that didn't land after 10 minutes were removed"?
Line 192: it isn't clear what the experimental setup is for the noisy size cue. What was the training regime, what was the testing regime?
Line 256: what is 30% and 42% a percentage of?
Reviewer 2 Report
Comments:
- L13-18: This reviewer recommends that these lines be rewritten in past tense consistent with the concluding sentence (L18-19)
- L49-51: Since all three detrimental outcomes listed result from neonicotinoid exposure perhaps it would be better to list the outcomes then conclude the sentence with "when exposed to neonicotinoid pesticides"
- L71-74: This should be rephrased in the past tense
- L77-78: Please provide the physical location of Koppert Biological 77 Systems and Biobest Sustainable Crop Management
- L79, 146-150, 157-159: Please rephrase in past tense
- L81-83: Was this time period determined experimentally? If a precedent was set by previous studies consider citing them here
- L87-88: What was the sample size?
- L95: What are the major chemical components of this odor?
- L125: What was reasoning behind using Safer R© undiluted, was this according to the product label or to achieve a certain reference concentration?
- L162: What was the heating mat temperature?
- L243-260: Was Reliant the only fungicide tested for these experiments? If so, this is not clear in the methods as L113-131 implied that three fungicide formulations were tested.
- In general, the discussion lacks elaboration on the study findings and their implications. There is also no discussion of the underlying mechanisms by which fungicide exposure induced observed effects nor comparisons with other studies using similar methodology/similar findings other than a brief summary (L262-264).
- L275: Consider replacing "this manuscript" with "we"
- For Figure 3 the legend has MO color coded red but this is not reflected in the bar graph, according to Table 1 no bees responded incorrectly, thus MO can be removed from the legend for Fig. 3.
Round 2
Reviewer 1 Report
The authors have done a superb job in responding to prior review, dropping an unneeded part that was confusing and clarifying the objectives in the introduction. The discussion is also greatly improved. In particular, the current manuscript's adjusted presentation of these results juxtaposed with the Ryalls et al. paper at the end of the discussion is very insightful in appreciating the strengths and limitations of both papers, the current state of knowledge in this field, and the further scientific landscape ahead.
There were just 2 places where I struggled with wording: lines 118-119 (objective 1 in introduction) were too dense for me to parse in the first several readings but I eventually figured it out. Lines 336-339 of discussion: are there extra words or missing words? I never figured out what the phrase "are carbon characteristics" goes with.